# A versatile, high-efficiency platform for CRISPR-based gene activation

Amy J. Heidersbach [1,3] ✉, Kristel M. Dorighi[1,3], Javier A. Gomez[2], Ashley M. Jacobi [2] & Benjamin Haley [1] ✉

CRISPR-mediated transcriptional activation (CRISPRa) is a powerful technology for inducing gene expression from endogenous loci with exciting applications in high throughput gain-of-function genomic screens and the engineering of cell-based models. However, current strategies for generating potent, stable, CRISPRa-competent cell lines present limitations for the broad utility of this approach. Here, we provide a high-efficiency, self-selecting CRISPRa enrichment strategy, which combined with piggyBac transposon technology enables rapid production of CRISPRa-ready cell populations compatible with a variety of downstream assays. We complement this with an optimized guide RNA scaffold that significantly enhances CRISPRa functionality. Finally, we describe a synthetic guide RNA tool set that enables transient, population-wide gene activation when used with the self-selecting CRISPRa system. Taken together, this versatile platform greatly enhances the potential for CRISPRa across a wide variety of cellular contexts.

Recent advances in genome engineering technology have enabled unprecedented opportunities for exploring the consequences of altered gene function or expression in a variety of model systems[1,2]. Driving many of these efforts has been the adaptation of the microbial CRISPR/Cas9 system for use in eukaryotic organisms[3]. Cas9's defining feature, as an easily programmable RNA-directed double-stranded DNA (dsDNA) nuclease, has inspired the creation of genome-scale perturbation libraries and subsequent loss-of-function screens across hundreds of human cell lines[4–6]. These screens have proven invaluable for uncovering genotype and cell lineage-specific gene dependencies, which continue to inform basic as well as clinical research efforts[7].

Cas9 can also be engineered for expanded use beyond the creation of targeted dsDNA breaks. The fusion of transcriptional repressor or activator domains to a nuclease-dead form of Cas9 (dCas9) enables CRISPR-mediated transcriptional interference (CRISPRi) or activation (CRISPRa), respectively[8–10]. As such, CRISPRa is a compelling technology for the activation of endogenous gene expression in disease models or gain-of-function screens[11–13]. A host of activator domains and transgene expression systems have been engineered to enable the production of CRISPRa-competent cells[14]. However, current strategies for engineering CRISPRa-transgenic cell lines are inefficient, prone to silencing, and often necessitate a labor-intensive single-cell cloning process. Gene and cell line-dependent variability pose further limits on the scalability of CRISPRa.

Here, we provide a comprehensive platform based on the Synergistic Activation Mediator (SAM)[13] CRISPRa concept, that takes advantage of a self-selection mechanism to create uniform, potent, and stable CRISPRa-competent cell populations without the need for clonal selection. In addition, we demonstrate the effectiveness of an optimized SAM-compatible single-guide RNA (sgRNA) variant that both improves the function of suboptimal sgRNAs and enables activity from sgRNAs found to be inactive with earlier-generation scaffolds. We show that this alternative sgRNA format is not only capable of facilitating stable gene expression, but that it can also be used for transient target activation through a chemically synthesized guide RNA tool set. Altogether, this user-friendly platform maximizes the potential for CRISPRa across a breadth of cell-based contexts and genetic loci.

[1]Department of Molecular Biology, Genentech Inc., South San Francisco, CA, USA. [2]Integrated DNA Technology Inc, Coralville, IA, USA. [3]These authors contributed equally: Amy J. Heidersbach, Kristel M. Dorighi. ✉e-mail: heidersbach.amy@gene.com; haley.benjamin@gene.com

## Results

### A self-selecting CRISPRa strategy for the rapid generation of stable, high-efficiency CRISPRa cell populations

Several dCas9-activator concepts have been described[12]. In pilot experiments, we observed consistent evidence of target activation with the Synergistic Activation Mediator (SAM) system, and selected this platform for optimization studies. The SAM system poses a challenge, however, owing to the size and number of discrete elements that must be introduced in order to create a stable CRISPRa-ready cell population. These include a dCas9-VP64 fusion protein, an MCP-(MS2 coat protein) p65-HSF1 co-activator fusion protein (MPH), and any number of selection markers. Combined, these components and their associated regulatory sequences exceed the conventional limit for efficient lentiviral packaging[15], often necessitating a multi-vector delivery strategy[13,16]. The piggyBac transposon system[17], on the other hand, allows for both a higher cargo capacity and the incorporation of multiple transgene cassettes within a single vector. PiggyBac-based strategies have been utilized for CRISPRa-based cell line generation[18,19], and, similar to lentivirus, its use results in random genomic integration and functional heterogeneity within the cell population. The resulting low efficiency populations are often incompatible with demanding applications like functional genomic screening without the further

derivation and characterization of highly active clones. Due to the laborious and time-consuming nature of this process we aimed to develop a simple and efficient bulk selection method to enrich for stable, uniform, and potent CRISPRa-ready cell populations.

To this end, we designed a series of multi-component CRISPRa piggyBac vectors which employed individual selection strategies for the enrichment of transgenic cells (Fig. 1a). In each context, expression of the CRISPRa activator elements was driven by a human EF1α promoter, and this was complemented by a distinct mechanism for the transcription of a co-expressed Puromycin resistance gene (Puro[r]). Similar to previous studies, we created a dual promoter selection vector[19] (Fig. 1a-top row) where Puro[r] was driven by an independent promoter (PGK) and a single transcript vector[20] where Puro[r] was transcriptionally linked to the CRISPRa machinery (Fig. 1a-middle row). The theoretical selection pressure exerted by these strategies should be on maintaining transgene genomic integration in the case of the dual promoter vector and on sustained transgene expression in the case of the single transcript system (Fig. 1a-right column).

As a readout for CRISPRa function, we also incorporated a GFP reporter downstream of a self-activating (SA) promoter sequence. Here, a minimal promoter was derived from the TRE3G promoter (Takara Bio), in which the tet operator array has been replaced by a single 19 bp

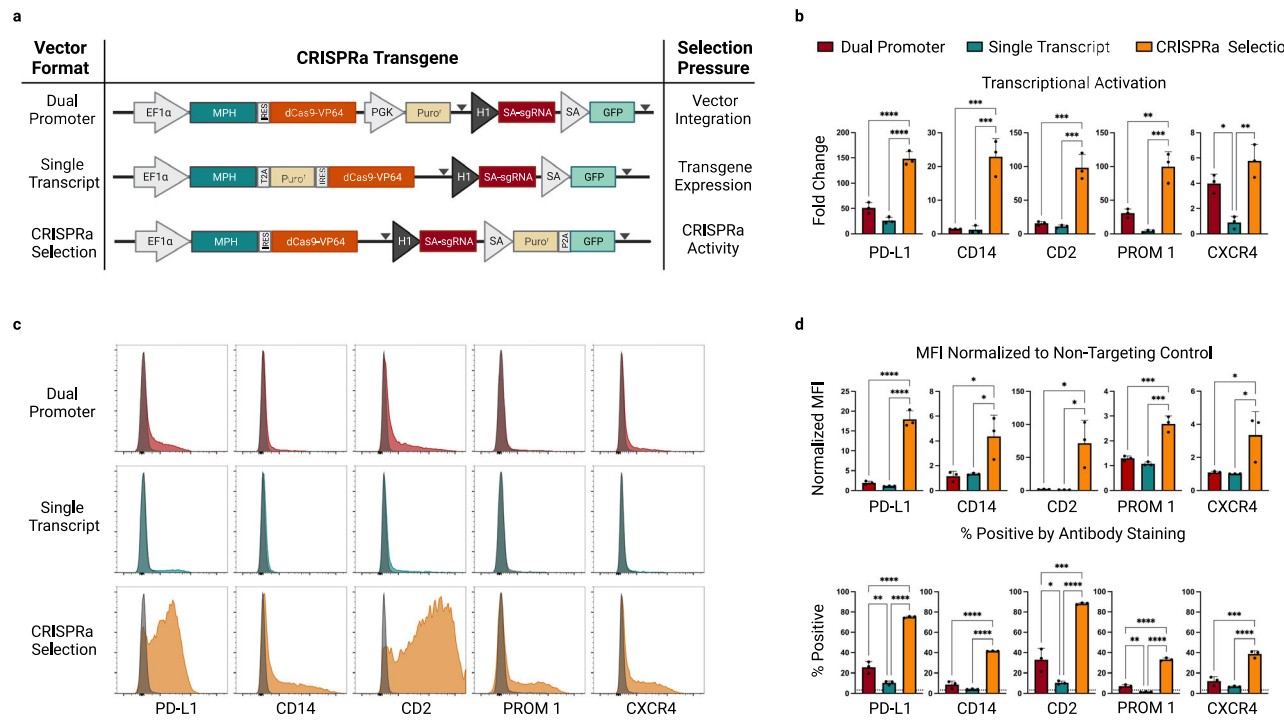

**Fig. 1 | A self-selecting CRISPRa piggyBac vector for the rapid generation of stable, high-efficiency CRISPRa cell populations. a** Vector format and selective strategy for the evaluated piggyBac CRISPRa expression-reporter vectors. Expression of the MCP-P65-HSF1 (MPH) activator and dCas9-VP64 is driven by a constitutive, human EF1α promoter. A human H1 promoter drives constitutive expression of a sgRNA complementary to the self-activating (SA) promoter upstream of a GFP reporter. Expression of a Puro[r] gene is driven either by its own constitutive promoter (dual promoter), transcriptionally linked to the MPH/dCas9-VP64 (single transcript) or under control of the CRISPRa dependent SA promoter (CRISPRa selection). Gray triangles indicate the location of LoxP sites. PiggyBac engineered K562 populations were generated in triplicate (*n* = 3 biologically independent samples) for each vector format and enriched with Puromycin selection. sgRNAs complementary to the promoter proximal region of the indicated genes were cloned into a lentiviral vector context containing a mTagBFP2/Zeocin selection cassette (Supplementary Fig. 1b). Following transduction and Zeocin selection target gene expression was evaluated by quantitative RT-PCR (qRT-PCR) (**b**), and

flow cytometry at day 14 post-infection (**c**, **d**). Representative histograms for each condition are overlaid with histograms from stained cell populations expressing a non-targeting control sgRNA (**c**) (gray). Infections were performed in duplicate (*n* = 2 technical replicates per biological replicate) and averaged. (Median fluorescence intensity (MFI) was normalized to MFI of an antibody-stained sample expressing a non-targeting sgRNA (**d**, top). Percentage of cells positive by antibody staining is presented (**d**, bottom) and background staining from a control sample expressing a non-targeting sgRNA is indicated with a dashed horizontal line for each gene. Data are presented as mean values +/− SD. Statistical comparison was performed by an unpaired one-way ANOVA adjusted for multiple comparisons. *$p < 0.05$, **$p < 0.01$, ***$p < 0.001$, ****$p < 0.0001$. EF1α -Elongation factor alpha, GFP-green fluorescent protein, dCas9-vp64-nuclease dead spCas9 + vp64 activator fusion, P2A-porcine teschovirus-1 2A self-cleaving peptide, HSF-heat shock factor, PD-L1-Programmed death-ligand1 (CD274), CD14-cluster of differentiation 14, CD2-Cluster of differentiation 2, Prom1-prominin-1 (CD133), CXCR4-C-X-C chemokine receptor type 4 (CD184). Source data are provided as a Source Data file.

sequence in context with a GGG protospacer-adjacent motif (PAM) in order to create a Cas9 targeting site. This target site can be recognized by a constitutive, H1 promoter-driven, self-activating sgRNA (SA sgRNA) that was designed to avoid association with endogenous human or mouse genomic sequences. Accordingly, in the presence of functional CRISPRa machinery, the SA sgRNA will activate the expression of the co-encoded GFP reporter. Building on the SA concept, we devised a third strategy, which we term CRISPRa selection (CRISPRa-sel), where the SA promoter is configured to express a T2A-linked Puroʳ and GFP reporter cassette (Fig. 1a-bottom row, Supplementary Fig. 1a).

We evaluated the relative efficiency of each selection strategy in the human K562 cell line. Here, CRISPRa vectors were electroporated, and following a minimum 5-day expansion period populations were selected with Puromycin for a minimum of 1 week to remove cells that had not integrated the piggyBac vectors. Selected populations were subsequently transduced with lentiviral vectors (Supplementary Fig. 1b) expressing SAM-compatible sgRNAs targeting the promoter-proximal regions of five cell surface receptor genes (*PD-L1*, *CD14*, *CD2*, *PROM1/CD133*, *CXCR4*). Gene expression analysis was performed a minimum of 10 days following the introduction of each sgRNA and 1 week in the presence of Zeocin. Quantitative RT-PCR (qRT-PCR) data from each condition revealed consistently improved gene activation with the CRISPRa-sel system relative to the other formats (Fig. 1b). We further used flow cytometry to quantitatively assess cell surface protein expression on an individual cell level (Fig. 1c, d; Supplementary Fig. 1c). This analysis demonstrated a dramatic enhancement in gene activation with the CRISPRa-sel system both in terms of absolute protein expression, by way of normalized median fluorescence intensity (MFI), and percent positive-stained cells. While the dual promoter and single transcript systems produced populations with only a small fraction of cells capable of inducing endogenous gene expression, the CRISPRa-sel strategy resulted in a high proportion of active cells, in some cases achieving near population-wide activation (i.e. *PD-L1* and *CD2*). To confirm the broad applicability of our findings, we expanded our analysis to two additional, unrelated human cell lines (Supplementary Fig. 1d, e) and observed similar trends.

In addition to endogenous target activation, we evaluated whether our integrated CRISPRa-dependent GFP reporter was effective for identifying CRISPRa-competent cells. To our surprise, we found that GFP intensity did not reliably correlate with endogenous target gene activation across the tested selection formats and cell lines (Supplementary Fig. 2). While correlations were generally higher in the dual promoter and single transcript formats ($R = 0.56-0.84$ and $R = 0.40-0.86$, respectively), correlations between endogenous gene activation and GFP reporter expression were poor in the CRISPRa-sel format ($R = 0.27-0.39$) (Supplementary Fig. 2a). To further evaluate the relationship between GFP expression and endogenous gene activation in the CRISPRa-sel context, we expanded our analysis to a second endogenous target gene (*CD2*) (Supplementary Fig. 2b) and observed similarly weak correlations ($R = 0.35-0.38$).

Despite this observation when analyzed in bulk, we wanted to determine if GFP expression could be used to facilitate the isolation of high-functioning single cell clones. We engineered the CRISPRa-sel system into four unrelated cell lines and following Puromycin selection we sorted cell populations based on high, medium, or low GFP expression via fluorescence-activated cell sorting (FACS) (Supplementary Fig. 2c, d). From these sorted populations, single-cell clonal lines were derived, and upon expansion were transduced with sgRNAs targeting distinct endogenous genes (*PD-L1*, *CXCR4*) or a non-targeting control. Interestingly, while relative GFP expression levels were maintained in the clones post-expansion (Supplementary Fig. 2c-left), there was no clear relationship between reporter expression and endogenous target activation in three of four cell lines evaluated (Supplementary Fig. 2c-center/right and Supplementary Fig. 2d). These data suggest that, in the context of the CRISPRa-sel system, selection with a

CRISPRa-dependent fluorescent reporter is not a broadly applicable strategy for further enrichment of CRISPRa-competent populations, beyond what is achieved with Puromycin selection. While other groups have reported successful enrichment with fluorescent CRISPRa responsive reporters[20], our data suggest such strategies are potentially more useful in the context of low efficiency systems like the dual promoter or single transcript formats where the number of active cells in the population is low and the functional difference between active and inactive cells is high. On the other hand, GFP-based reporters may not be sensitive enough to discriminate effectively between cells within more uniform CRISPRa-sel-derived populations. Therefore, we focused on antibiotic-selected populations for the remainder of our platform optimization efforts.

## SAM guide RNA scaffold optimization for enhanced CRISPRa activity

Subtle changes in scaffold sequence and structure have been shown to affect guide RNA function[13,16,21,22] and we reasoned that the conventional SAM-2.0 scaffold could be re-engineered to improve activity. The MPH activator utilized by the SAM system binds to two separate MS2 aptamers within the SAM-2.0 sgRNA; one in the tetraloop and one in stemloop two (Supplementary Fig. 3a). Focusing on the tetraloop, we used rational design to create several distinct SAM-compatible scaffold variants (Supplementary Fig. 3a, b, Fig. 2a). Previous reports have indicated that Pol-III-based guide expression can be enhanced by removing a polyU tract in the tetraloop, which can serve as a premature transcriptional termination sequence[21,22] (Supplementary Fig. 3b [GNE-1]). In addition, we hypothesized that increasing the stability or accessibility of the MS2 aptamer segment within the tetraloop could encourage greater associations with MPH complexes, further improving CRISPRa efficiency. To explore these possibilities, we coupled polyU elimination with an alternate, GC-rich stem extension sequence proximal to the MS2 aptamer (Supplementary Fig. 3 [GNE-2])[21]. Finally, we combined both stem extension features with the removal of a bulge sequence directly adjacent to the MS2 aptamer (Supplementary Fig. 3 [GNE-3]).

To evaluate the relative efficiency of these scaffolds, we lentivirally transduced CRISPRa-sel-engineered K562 populations with sgRNAs targeting three endogenous genes (*PD-L1*, *CD14*, or *KDR*) in either the SAM-2.0 scaffold format or one of our three unique variants (Supplementary Fig. 3c). By flow cytometry, higher target expression was observed with several of the alternative scaffold variants, but GNE-3 showed the most consistent improvement over 2.0, both in terms of gene product levels (normalized MFI) and the percentage of activated cells across the population. We subsequently expanded our comparison of the 2.0 and GNE-3 scaffolds to include six cell surface receptor genes, using five unique sgRNAs per gene, to account for gene and spacer-specific variability. Analysis of target transcript (qRT-PCR) and protein (flow cytometry) expression (Fig. 2b, c; Supplementary Fig. 4a, b) revealed a broad enhancement of target activation with the GNE-3 scaffold versus the 2.0 backbone, with several sequences achieving between 5–10-fold improved gene induction with the GNE-3 variant. To confirm that the GNE-3 scaffold was beneficial in other cell contexts, we expanded our analysis to two additional cell lines. As before, we found activation of *PD-L1*, as measured by cell surface staining in 293T and Jurkat cells, (Supplementary Fig. 4c) was consistently higher with the GNE-3 scaffold. Taken together these data suggest that the GNE-3 scaffold improves both the breadth and magnitude of gene activation across a variety of spacer, target and cellular contexts. In addition, we found that relative target gene activation was largely consistent when comparing transcript level or cell surface protein stain (Supplementary Fig. 4d) for most targets, and therefore chose to move forward with validated flow cytometry assays for subsequent experiments owing to the quantitative nature of this assay at both the population and individual cell levels.

**Fig. 2 | Relative CRISPR activation efficiency of sgRNAs with an optimized MS2 aptamer-containing scaffold. a** Structure diagram of the MS2-aptamer containing tetraloop in the 2.0 sgRNA format[13] (left) or an optimized tetraloop structure (right and Supplementary Fig. 3). The optimized GNE-3 tetraloop contains an additional stem extension and removal of a polyU tract[21]. In addition, the bulge region connecting the MS2 aptamer and stem extension region 1 in the 2.0 format has been removed. **b** Flow cytometric analysis of target gene activation by sgRNAs with either a 2.0 (orange) or GNE-3 (teal) scaffold context. Representative histograms of analyzed K562 CRISPRa-sel populations infected with five distinct spacer sequences targeting the promoter proximal region of *PD-L1* (top) or *CD2* (bottom).

Populations infected with a non-targeting sgRNA sequence overlaid (gray). **c** Activation of 6 target genes by GNE-3 sgRNAs normalized to the activation efficiency of the same spacer sequence in a 2.0 format (dashed line). Normalized gene activation was evaluated in Zeocin selected populations by qRT-PCR (left) at day 14 post-sgRNA infection or by flow cytometry (right) at day 10 post-sgRNA infection. Data are presented as mean values +/− SD. *n* = 4 independent biological replicates per sgRNA. # indicates an inactive spacer where neither 2.0 nor GNE-3 sgRNAs activated the target gene (cutoff <1.5-fold over control). Source data are provided as a Source Data file.

## CRISPRa-sel promoter optimization and evaluation in a panel of human cell lines

While the combination of our CRISPRa-sel system with the GNE-3 scaffold demonstrated improvement in overall CRISPRa efficiency, we continued to observe variable target activation across cell lines (Fig. 3a–c [EF1α], Supplementary Fig. 5 [EF1α]). The strength of Pol-II promoters, which drive expression of the CRISPRa machinery (Fig. 3a), can differ dramatically across cell types[23] potentially contributing to the context-dependent efficacy of CRISPRa. To evaluate how promoter use impacts the efficiency of the CRISPRa-sel system, we engineered a panel of three cell lines (K562, 293T and Jurkat) with the original EF1α-

based CRISPRa-sel vector or versions that incorporated three distinct cytomegalovirus (CMV)-derived Pol-II promoter variants (CBh, CMV, and CAG) (Fig. 3a–c; Supplementary Fig. 5) to drive expression of the activator machinery.

Attempts to engineer CRISPRa-sel populations were successful in all but one cell line context (Jurkat + CMV-CRISPRa-sel) (Fig. 3b), in which only a low number of slow-growing clones were recovered following Puromycin selection. To evaluate the relative efficacy of each promoter, populations were transduced with GNE-3 sgRNAs targeting *PD-L1* or *CD2*. Unlike the more heterogeneous activation observed with the EF1α, CBh, and CMV promoters, the CAG promoter induced

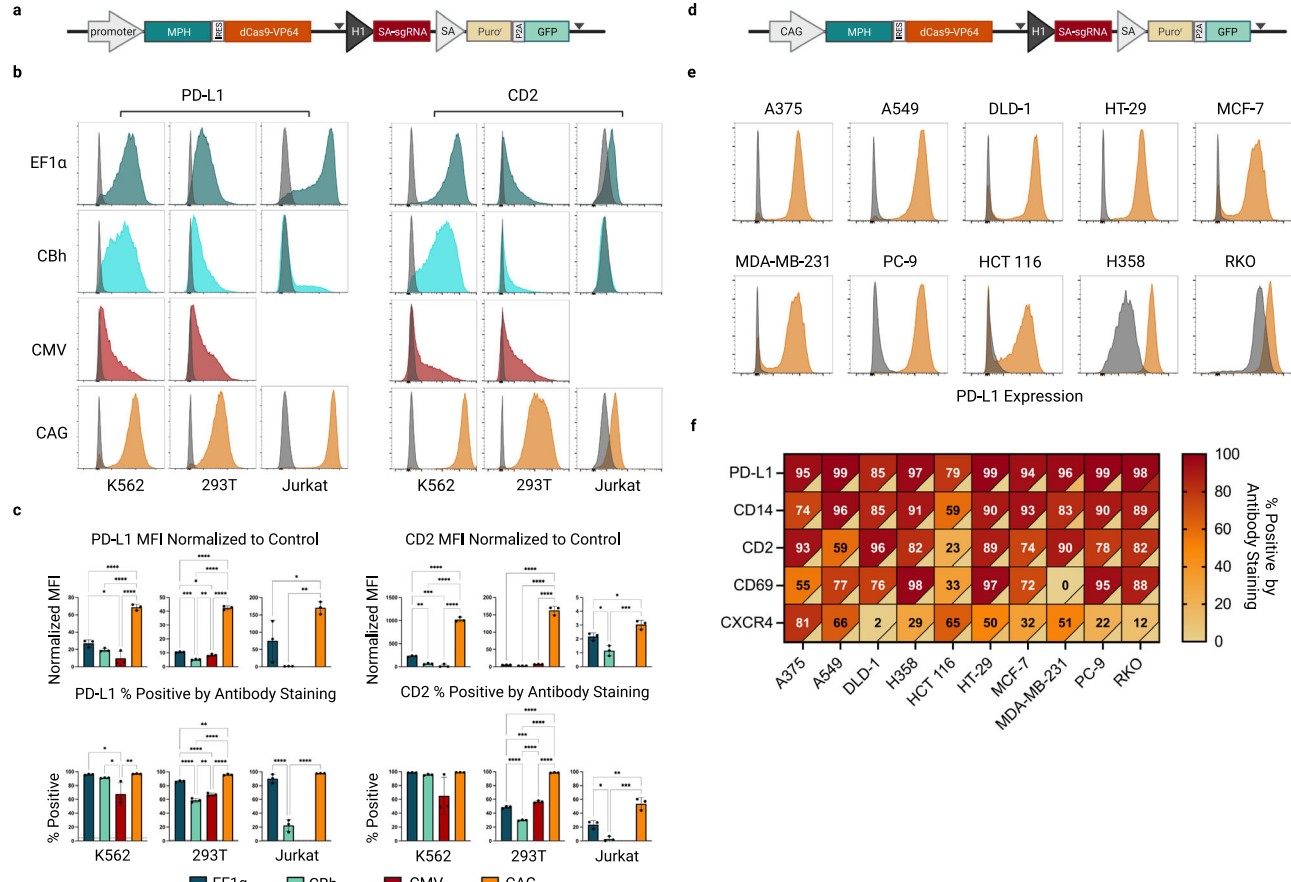

**Fig. 3 | Promoter optimization and application of the CRISPRa-sel strategy across a panel of commonly used cell lines. a** Schematic representation of the CRISPRa-sel vector indicating the location of the promoter driving expression of the MPH/dCas9-VP64 transcript. **b, c** Activation of *PD-L1* (left) or *CD2* (right) target genes evaluated by flow cytometry in K562, 293T and Jurkat cell lines engineered with CRISPRa-sel piggyBac vectors utilizing an EF1α (teal), CBh (aqua), CMV (maroon) or CAG (orange) promoter 14 days post-infection with a GNE-3 sgRNA. **b** Activation displayed by representative histograms overlaid with expression profiles from cells infected with a non-targeting sgRNA (gray). **c** Normalized median fluorescence intensity (MFI) (top) or percentage positive (bottom) of indicated genes/cell populations by antibody staining. The percent positive of stained control populations infected with a non-targeting sgRNA are indicated by a dashed horizontal line. (Note: CMV CRISPRa-sel Jurkat populations did not grow out efficiently and were not included in the analysis.) **d** Schematic representation of the CAG CRISPRa-sel piggyBac vector. **e** Representative flow cytometric histograms of *PD-L1* activation across 10 commonly used cell lines engineered with a CAG-driven

CRISPRa-sel piggyBac vector and *PD-L1* targeting GNE-3 sgRNA. **f** Heatmap representing the percent positive of five target genes (*PD-L1, CD14, CD2, CD69* and *CXCR4*) across 10 CAG CRISPRa-sel engineered cell lines (A375, A549, DLD-1, H358, HCT 116, HT-29, MCF-7, MDA-MB-231, PC-9 or RKO). Percent positive of stained cell populations expressing a non-targeting sgRNA represented colorimetrically in the lower right corner of each cell. Gene-activating or control guides were expressed using dual sgRNA lentivectors (Supplementary Data 1). CRISPRa cell populations generated in triplicate (*n* = 3 biologically independent samples) and infected with indicated sgRNAs in technical duplicates (*n* = 2 technical replicates per biological sample) which were averaged before statistical comparison was performed by an unpaired one-way ANOVA adjusted for multiple comparisons. Data are presented as mean values +/− SD. *$p < 0.05$, **$p < 0.01$, ***$p < 0.001$, ****$p < 0.0001$. Gray triangles indicate the location of LoxP sites. CBh -Chicken β-actin hybrid promoter, CMV- human cytomegalovirus immediate-early gene enhancer/promoter or CAG-CMV enhancer-chicken β-actin-rabbit β-globin synthetic hybrid promoter. Source data are provided as a Source Data file.

distinctly uniform and potent gene expression for each of the tested cell lines and targets (Fig. 3b, c). We expanded our assessment to include three additional endogenous targets (*CD14*, *CXCR4*, and *CD69*) and saw comparable results (Supplementary Fig. 5a, b). Importantly, this demonstrated that population-wide CRISPRa was achievable with limited cell culture manipulation beyond bulk antibiotic selection.

Our design of the CRISPRa-sel vector includes loxP sites flanking the SA sgRNA/selection cassette to enable its removal. While the presence of the SA cassette is necessary to drive selection marker expression, the SA sgRNA may compete to some extent with gene-targeting sgRNAs for dCas9 binding. In theory, this could impact the overall efficiency of target activation. To better understand how the presence of the SA cassette influences the efficiency of endogenous gene activation, we performed a series of experiments in cell lines with the intact SA cassette or following its removal with Cre recombinase post-selection (Supplementary Fig. 6). To this end, stable CAG-

CRISPRa-sel K562 or 293T cell lines were transduced with gene-targeting sgRNAs (*PD-L1*, *CD2*, or *CD14*) (Supplementary Fig. 6a) and selected with Zeocin to remove uninfected cells. Populations were then divided and either maintained in culture (mock) or treated with an mRNA encoding Cre. After 7 days, cells were evaluated by flow cytometry. Loss of the SA cassette was confirmed by the significant reduction in GFP expression in the Cre-treated populations (Supplementary Fig. 6b). Under these conditions no consistent difference in target gene activation was observed between the Cre-treated or untreated populations, suggesting competition from the SA sgRNA has minimal impact on target gene activation (Supplementary Fig. 6c, d). Furthermore, these data suggest that in short term studies, the SA cassette may not be essential for maintaining CRISPRa populations after the initial selection process.

In order to confirm the broad utility of the CAG-CRISPRa-sel and GNE-3-sgRNA system, we engineered an additional panel of ten

commonly used cell lines (Fig. 3d–f). After bulk selection of the CAG-CRISPRa-sel transgenic cell lines, introduction of a *PD-L1*-specific sgRNA led to strong, uniform target induction (-79–99% of the cell population) (Fig. 3e, f). We then expanded this analysis to four additional target genes per cell line, and while we observed some context-dependent variability for individual genes, robust activation in ≥75% of the cell population was seen in the majority of conditions (Fig. 3f). Notably, beyond activating genes with little or no background expression, we were able to induce population-wide upregulation of genes with high basal expression (Fig. 3e [H358[24]],[RKO[25]]). Taken together, these data indicate that the CAG-CRISPRa-sel system in conjunction with the GNE-3 scaffold greatly expands the utility of stable CRISPRa across a breadth of cell backgrounds and target genes.

### Optimized, multi-format synthetic guide RNAs for transient CRISPRa

Synthetic guide RNAs can be generated quickly and have proven effective for Cas9-mediated gene disruption purposes ranging from the creation of in vitro and in vivo models to arrayed genetic screens[26]. While synthetic gRNAs have previously been applied in the context of CRISPRa[27], thus far they have not been widely adopted possibly due to their low efficiency with suboptimal CRISPRa systems and the challenges associated with synthesizing long sgRNAs. Until recently, dual MS2 aptamer-containing sgRNAs, like the -160 nucleotide GNE-3 spacer sequence and scaffold, exceeded the length of reliable direct synthesis methodology[28]. As an alternative approach, the use of easier-to-synthesize two-part gRNAs (crRNA + tracrRNA scaffold) is an attractive possibility. The design of these guides, however, must allow for efficient strand annealing while maintaining the structure of the MS2 aptamer loops[26]. In addition, any synthetic guide RNA, regardless of format, needs to be stable enough throughout delivery, dCas9 association and target binding to induce measurable gene activation. Given recent advances in RNA synthesis and chemical stabilization, and to yet further expand the utility of CRISPRa, we set out to develop an optimized GNE-3-based synthetic gRNA platform.

To evaluate the impact of chemical modifications on the efficiency of CRISPRa induced by transient delivery of synthetic guides in cultured cells, we synthesized a set of sgRNAs based on the GNE-3 scaffold targeting four endogenous genes (*PD-L1, CD14, CD2, CXCR4*) with or without modified stabilizing nucleotides[29]. Individual unmodified sgRNAs were compared to identical sgRNAs containing three terminal phosphorothioated 2′ O-methyl ribonucleotides at both the 5′ and 3′ ends (Fig. 4a). Three days after electroporation into a CAG-CRISPRa-sel-engineered K562 population, we observed clear evidence of gene activation. We found that the modified sgRNAs demonstrated a clear advantage over the unmodified guides across all targets evaluated (Fig. 4b, c). To our surprise, activation with the transient modified synthetic sgRNAs was qualitatively similar in some cases to stable sgRNA expression, with near-population-wide expression achieved for two of four target genes.

We next sought to determine if the GNE-3 sgRNA variant also outperformed the 2.0 scaffold in a synthetic context. To this end we generated identical end-modified sgRNAs for the 2.0 variant. Direct comparison in the CAG-CRISPR-sel K562 model demonstrated a general trend towards higher activation with the GNE-3 sgRNAs, although the differential was somewhat reduced compared to the stable sgRNA context (Supplementary Fig. 7).

User accessibility of synthetic guide RNA-mediated CRISPRa could be enhanced by lowering the cost and technical skill required for reagent synthesis. In principle, this could be achieved by minimizing the length of the guide RNA segments with a more native, annealed two-part crRNA-tracRNA format. In order to create synthetic material that permitted crRNA and tracrRNA hybridization while maintaining the GNE-3 scaffold loop structure, we developed two distinct concepts (Supplementary Fig. 8a). In Format 1, strand 1 includes the spacer

sequence and a segment of the GNE-3 MS2 containing tetraloop (Supplementary Fig. 8a-teal), which anneals to strand 2 containing the final portion of the tetraloop as well as stemloop 1, stemloop 2 (with the second MS2 aptamer) and stemloop 3. Separately, in Format 2, strand 1 exclusively comprises the spacer plus a short region (Supplementary Fig. 8a-orange) with complementarity to strand 2. Strand 2 of this format encodes the majority of the tetraloop and stemloops 1–3. All RNA oligonucleotides contain 5′ and 3′ stabilizing modifications similar to our optimized synthetic GNE-3 sgRNA. We incorporated identical spacer sequences within both formats and evaluated their relative effectiveness for activating four separate genes within CAG-CRISPRa-sel K562 cells. When we analyzed target activation by flow cytometry three days post-electroporation we saw higher gene activation with Format 1 across all targets (Supplementary Fig. 8b–c), and this format became a focus for follow-up studies.

Recently, a two-part, SAM-compatible guide RNA system has been described and made commercially available[27]. Unlike the GNE-3 guide RNAs described herein, the commercial product contains fewer phosphorothioated 2′ O-methyl ribonucleotides and has only a single MS2-modified element within stemloop 2 (the MS2 sequence within the tetraloop being absent) (Fig. 5a-top). In order to evaluate the relative functionality of these synthetic guide RNAs, we compared GNE-3 sgRNAs and Format 1 two-part guide RNAs to the commercially available synthetic guide RNA format (1X MS2 two-part) in CAG-CRISPRa-sel-engineered K562 and 293T populations (Fig. 5b–e and Supplementary Fig. 9). We found that the GNE-3 sgRNA and two-part formats generally outperformed the single MS2 containing guide, with the GNE-3 sgRNA format providing the most consistent and potent activation across all tested contexts. The differential activity across guides was particularly pronounced in lower activity conditions (Fig. 5b, c, Supplementary Fig. 9a, b gRNA-1). Only under circumstances of high CRISPRa activity, such as in 293T cells, could measurable induction be achieved with all of the evaluated 2-part and sgRNA variants (Fig. 5d, e, gRNA-3/gRNA-4, Supplementary Fig. 9).

## Discussion

The potential for any genome engineering technology is limited by the breadth of cell types and loci for which it can be applied. By incorporating a self-selecting transgenic approach with enhanced SAM-compatible guide RNA scaffolds, we have demonstrated that robust, population-wide CRISPRa is achievable across a diverse panel of target genes and cell lines, all with minimal cell manipulation steps. In addition, we show that synthetic guide RNAs can be employed for highly efficient, short-term gene activation, in some cases with population-wide efficacy. While our described CRISPRa toolkit showed robust activity across multiple contexts, we identified several notable features of our system that present opportunities for iteration and improvement.

In our initial experiments we observed that while antibiotic selection with our SA system enabled generation of relatively uniform populations of CRISPRa-competent cells, further enrichment based on high SA GFP reporter expression did not result in cell populations with correspondingly robust endogenous gene activation. There are several possible explanations for this finding. Variability in the expression of the endogenous gene-targeting guide RNA due to positional effects of vector insertion could result in the decoupling of endogenous gene activation and SA GFP reporter expression. Separately, the GFP reporter in the SA context, unlike that in the dual promoter and single transcript systems, is transcriptionally linked to Puro[r]. Since we initially selected with Puromycin, it is possible that the GFP in this configuration is acting as a passive marker, the expression of which is determined primarily by the selective requirement of the co-transcribed Puro[r] and secondarily by the level of CRISPR-mediated gene activation. Given that antibiotic selection alone proved sufficient for robust CRISPRa enrichment, we have modified the SA cassette to create a

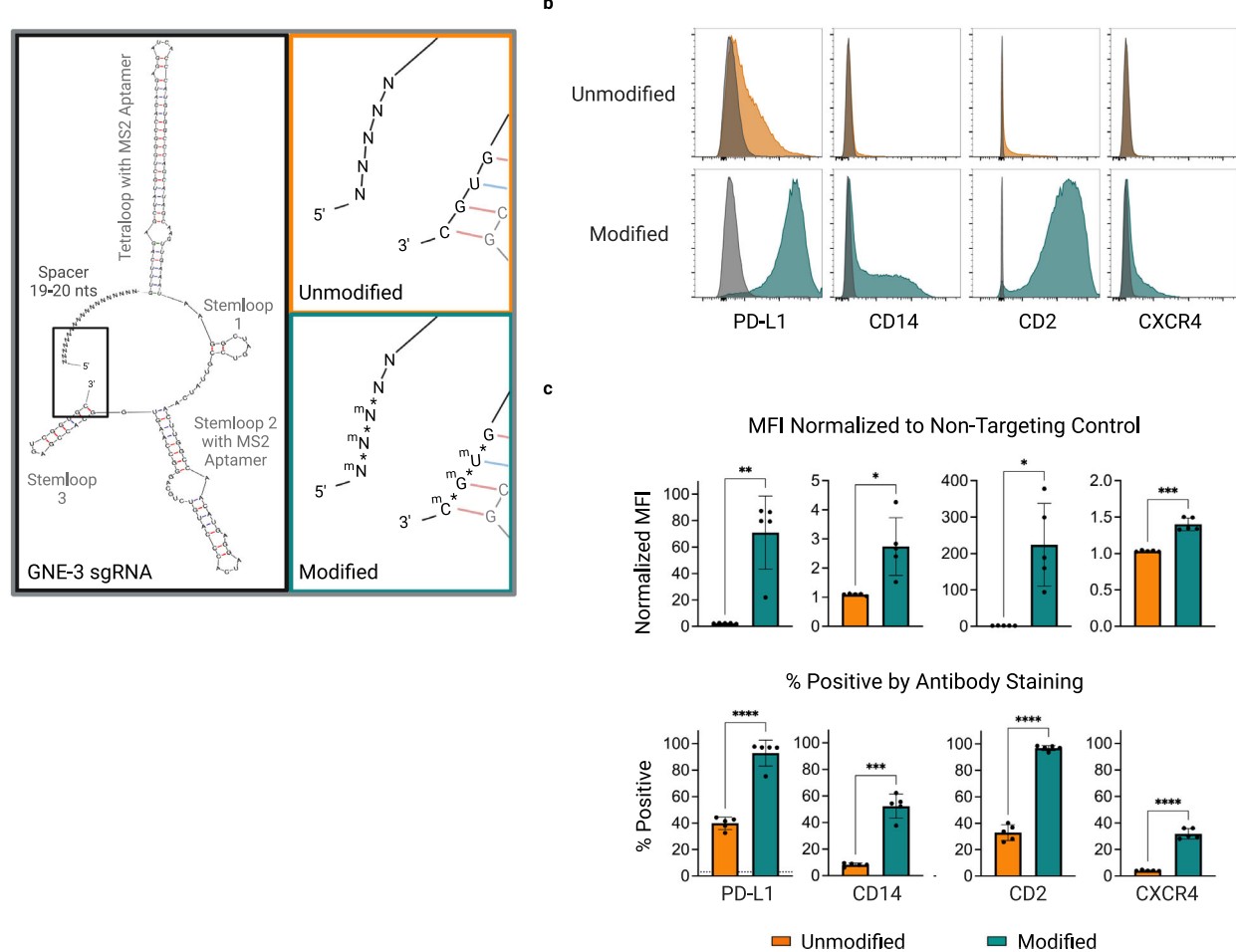

**Fig. 4 | Chemical modification of synthetic GNE-3 sgRNAs enhances target gene activation. a** Structural diagram of a full sgRNA with GNE-3 scaffold highlighting 5′/3′ ends (black box). Magnified view of sgRNA 5′/3′ end regions highlighting unmodified (orange) or modified (teal) nucleotides. Modified sgRNAs contain 2′-O-methyl (m)/ phosphorothioate (*) linker modifications. **b, c** Assessment of CRISPR mediated gene activation by unmodified or modified sgRNAs in a CAG-CRISPRa-sel engineered K562 population and assessed 3 days post-sgRNA delivery. **b** Gene expression displayed by representative flow cytometry histograms in populations electroporated with unmodified (top row, orange) or modified (bottom row, teal) GNE-3 sgRNAs. Stained cells electroporated with a non-targeting synthetic sgRNA overlaid in gray. **c** Median fluorescence intensity (MFI) of K562 populations stained with antibodies for the indicated genes (*PD-L1, CD14, CD2, CXCR4*) and normalized to a population of stained cells electroporated with a non-targeting sgRNA (top). Percentage of cells positive by antibody staining (bottom). Background staining of a cell population electroporated with a non-targeting control sgRNA indicated with a dashed horizontal line. *n* = 5 electroporation replicates per condition. Data are presented as mean values +/− SD. Statistical significance determined by an unpaired two-tailed *t*-test with a Welch's correction. *$p < 0.05$, **$p < 0.01$, ***$p < 0.001$, ****$p < 0.0001$. Source data are provided as a Source Data file.

series of antibiotic resistance-only vectors for increased versatility across models (Supplementary Data 1-full vector sequences).

Manipulation of the SA guide RNA and/or its expression may provide yet another avenue for future optimization. In our system, the SA guide expression context (H1 promoter combined with the 2.0 sgRNA scaffold) is suboptimal relative to the higher-efficiency U6 promoter and GNE-3 sgRNA employed for gene-targeting guides[30]. This may enable selection of more active CRISPRa populations, as only those cells which can activate the Puro[r] gene under such limiting conditions will survive. If this hypothesis is correct, then it is possible that reducing the functionality of our SA sgRNA further, for example, by employing a scaffold that lacks MS2 aptamers or uses a less efficient spacer sequence, could result in more robust selective pressure and, consequently, improved activity within the selected population.

Notably, we found that while the SA marker cassette was required for the initial selection process, it, along with the SA sgRNA, could be removed without substantial impact on CRISPRa efficiency in short-term studies. Although this may provide greater flexibility for downstream applications, by allowing researchers to subsequently introduce transgenes with the removed selection marker, we have not tested the effect of cassette deletion over the long-term. It remains possible that without maintenance of CRISPRa functionality through self-activated selection, the CRISPRa transgenes could be silenced or lose efficacy within cell populations, leading to greater variability during prolonged culture.

Advancements in gene activator technologies are inevitable, and we anticipate that self-selecting circuits will be compatible with future transcriptional and epigenetic modifier fusion proteins or extended Cas family member usage[1]. This will be critical for expanding the target space available for CRISPRa and for potentially enhancing gene expression at loci that show weak or modest induction with the SAM activator machinery. Despite the context-specific challenges particular cell lines or loci may pose, our data suggest that the SA concept coupled with the enhanced sgRNA scaffold and optimized synthetic guide RNAs will greatly improve the efficacy of CRISPR-mediated gene activation. Taken together, these versatile tools provide a robust platform to broadly enable future gain-of-function studies.

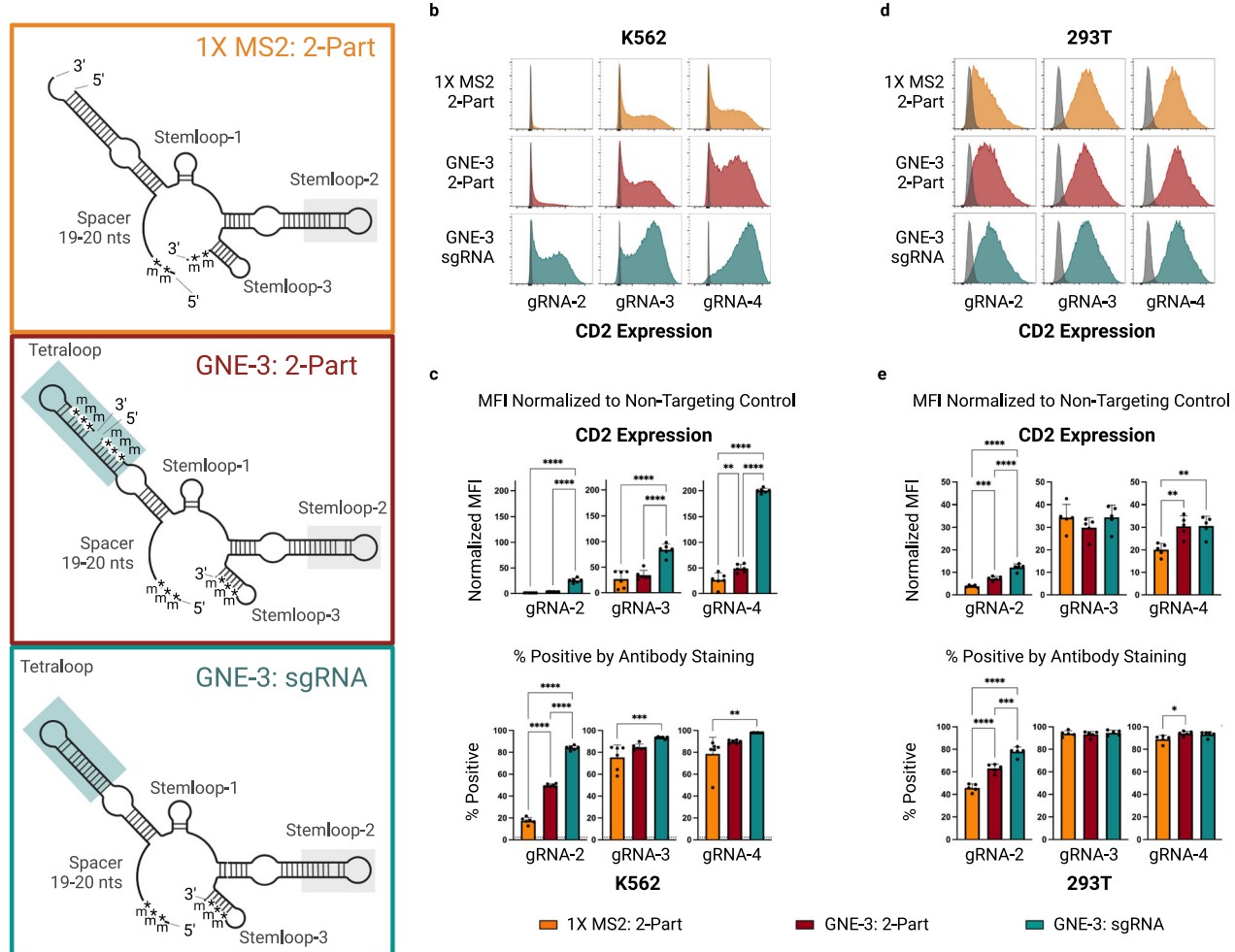

**Fig. 5 | Evaluation of CRISPRa synthetic guide formats across 2 cell lines.**
**a** Structure diagrams of a commercially available, chemically modified, 2-part synthetic gRNA containing a single MS2 aptamer loop (top-orange); a modified, 2-part Format 1 synthetic gRNA containing a GNE-3 scaffold (center-maroon); or a modified sgRNA with a GNE-3 scaffold (bottom-teal). Blue shaded boxes highlight the MS2 aptamer-containing GNE-3 tetraloop and gray shaded boxes indicate the MS2-apatmer on stemloop 2. **b–e** CRISPR-mediated transcriptional activation of a *CD2* target gene in two CAG-CRISPRa-sel-engineered cell lines (K562 or 293T) by electroporated modified, synthetic gRNAs in the formats depicted in **a**. *CD2* target activation by 3 spacer sequences in an engineered K562 cell line assessed by flow cytometry. *CD2* expression displayed by representative histograms overlaid with a control population (**b**) or summarized by median fluorescence intensity normalized to a non-targeting control (**c**-upper) or percent positive (**c**-lower). Percent positive of a stained control population treated with a non-targeting sgRNA are indicated by a dashed horizontal line. **d**, **e** *CD2* target activation by synthetic gRNA formats as in b-c but in a 293T cell line. Flow cytometry performed 3 days after synthetic guide delivery. Data are presented as mean values +/− SD. Statistical comparison between guide formats was performed by an unpaired one-way ANOVA adjusted for multiple comparisons. *$p < 0.05$, **$p < 0.01$, ***$p < 0.001$, ****$p < 0.0001$. $n = 6$ electroporation replicates for K562, $n = 5$ transfection replicates for 293T. m = 2'-O methyl. * = phosphorothioate linker. Source data are provided as a Source Data file.

## Methods

### Cell culture, electroporation, transfection
Cell line-specific culture and manipulation protocols described in Supplementary Data 1. All parental cell lines were sourced from the Genentech cell bank (gCell) where they were maintained under mycoplasma-free conditions and authenticated by STR profiling. FACS sorting and subsequent clonal derivation/analysis presented in Supplementary Fig. 2c, d was performed by WuXi AppTech.

### Lentiviral production/transduction
sgRNA-expressing and lentiviral packaging plasmids (VSVg/Delta8.9) were transiently co-transfected into 293T cells with Lipofectamine 2000. Lentiviral supernatants were harvested at 72 h and filtered through a 0.45 µm PES syringe filter (Millipore). Transduction with lentivirally encoded sgRNAs performed as described in Supplementary Data 1 with cell line-specific protocols. 3 days following lentiviral infection, cells were started on Zeocin selection at cell line-specific concentrations (Supplementary Data 1) in order to select for sgRNA-expressing cells. Prior to gene expression analysis, uniform selection of sgRNA-infected populations was confirmed by flow cytometric analysis of the co-expressed mTagBFP2 reporter.

### Flow cytometry
Antibody staining performed using manufacturers' recommended protocols and described in Supplementary Data 1. Data collection performed on BD Celesta, BD Fortessa or BD Symphony machines using FACSDIVA v8/v9 acquisition software. Data were subsequently analyzed by FlowJo2 v10.7/v10.8 (Becton, Dickinson & Co.). Gating strategy indicated in Supplementary Fig. 1c. Live cell populations were gated using FSC and SSC profiles. Where relevant, lentivirally transduced cells specifically were examined by gating on mTagBFP2 positive populations. If cell populations were selected to >95% mTagBFP2 positive then this gating step was omitted for some analyses. Populations were defined by gates established as indicated with 2 parameter

pseudocolor plots (Supplementary Fig. 1c or Supplementary Fig. 6b) with identical control cell lines expressing a non-targeting control guide RNA and stained/collected in parallel. Lentivirally transduced populations were analyzed a minimum of 10 days post-infection with the gene-targeting sgRNA. Experiment specific timing is described in the associated figure legends. Synthetic guide RNA experiments were analyzed at day 3 post-transfection or electroporation.

## qRT-PCR
Cell pellets for qRT-PCR analysis were collected 14 days post-lentiviral infection with a gene-targeting sgRNA, 11 days of which were in the presence of Zeocin to fully select for cells with genome-integrated guides. RNA extraction was performed with a Quick-RNA 96 well kit (Zymo). cDNA generation was performed with a high-capacity cDNA synthesis kit using random primers and RNase inhibitor (Thermo) following recommended protocols. Quantitative RT-PCR was performed with an ABI QuantStudio 7 Flex real time PCR system. Relative quantification/fold change (2^-ΔΔCT) analysis was performed by QuantStudio 7v2 software. A GAPDH control gene was used for normalization purposes.

## Synthetic gRNA electroporation/transfection
Direct synthesis and QC of the chemically modified sgRNAs and 2-part guide RNAs was performed by IDT (https://www.idtdna.com/pages). All synthetic gRNAs were resuspended in Nuclease-Free Duplex Buffer (30 mM HEPES, pH 7.5; 100 mM potassium acetate) (IDT). Commercially available modified, synthetic 2-part guide RNAs containing a single MS2 aptamer loop purchased from Horizon Inc. (https://horizondiscovery.com/). 2-part crRNA and tracrRNA oligonucleotides were combined at equimolar ratios prior to a denaturation/annealing protocol (95 °C for 5 min; cool to room temp 2°/s). sgRNAs were also treated with heat denaturation prior to use. Cell line specific synthetic guide delivery protocols detailed in Supplementary Data 1.

## RNA structure prediction
RNA folding performed using mFold[31] or bifold (https://rna.urmc.rochester.edu/RNAstructureWeb/Servers/bifold/bifold.html) algorithms.

## Figure production
Figure elements produced in Excel, Flowjo, and PRISM. Final figures created with BioRender.com.

## Statistics and reproducibility
Statistical tests performed as indicated in figure legends for each experiment. Error bars represent standard deviation from the mean. Data were analyzed using PRISM v9 (GraphPad Software, LLC) and/or Excel v16 (Microsoft) software. Bar plots/scatter plots and heatmaps were generated using PRISM.

## Reporting summary
Further information on research design is available in the Nature Portfolio Reporting Summary linked to this article.

## Data availability
The authors declare that processed data supporting the findings of this study are available within the article, its supplementary files, and Source Data file. Due to the large file size, the raw data is available on request from the corresponding author, requests will be answered in 2 weeks. Source data are provided with this paper.

## Materials availability
Biological materials will be provided to requesters through a material transfer agreement. Vector and guide RNA sequences are provided in Supplementary Data 1. Synthetic guide RNAs can be purchased through IDT.

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

## Acknowledgements
We would like to acknowledge Jean-Philippe Fortin, Colin Watanabe, Søren Warming, Anqi Zhu, Sandra Melo, Yassan Abdolazimi, Nadia Martinez-Martin, Clark Ho, Fabiola Juárez, and Letty Marroquin for thoughtful discussions and manuscript support.

## Author contributions
A.J.H., K.M.D., and B.H. conceived of and designed the study. A.J.H. and K.M.D. performed the experimental studies and carried out the data analysis. J.A.G. and A.M.J. provided synthetic gRNA reagents and reagent design support.

## Competing interests
A.J.H., K.M.D., and B.H. are full time employees of Genentech, Inc. and shareholders of Roche. Products and tools supplied by IDT are for research use only and not intended for diagnostic or therapeutic purposes. Purchaser and/or user is solely responsible for all decisions regarding the use of these products and any associated regulatory or legal obligations. J.A.G. and A.M.J. are employees of Integrated DNA Technologies, which offers reagents for sale similar to some of the compounds described in the manuscript.
