## [Peer Review File · Nature Communications]

Reviewers' Comments:

Reviewer #1:

Remarks to the Author:

Heidersbach and Dorighi et al. present a novel strategy for producing CRISPRa-transgenic cell lines using the SAM-system. The authors designed three vectors that rely on unique promoter arrangements to drive the expression of a puromycin (puroR) resistance cassette and delivered these to cells using a piggyBac transposon. They found that a selection strategy termed CRISPRa-selection (CRISPRa-sel), where a CRISPRa guide targets a self-activating promoter and drives the expression of both the puroR gene and GFP, resulted in greater levels of gene activation for a second guide targeting a gene of interest. Next, the authors altered the sgRNA scaffold by making three modifications to the MS2-aptamer-containing tetraloop and showed that this new structure, termed GNE-3, tended to increase CRISPRa sgRNA efficiency. The authors then compared the activity of four promoters in the CRISPRa-sel system and identified CAG as the top performer. Heidersbach and Dorighi et al. also optimized synthetic GNE-3 sgRNAs via chemical modifications and demonstrated their improved activation abilities in a transient CRISPRa setting. Finally, the authors generated a more accessible two-part synthetic GNE-3 gRNA system which outperforms commercially available alternatives.

Overall, we find that the authors support their claims thoroughly by comparing across a compelling number of cell lines and genes. The platform they present will be useful to the CRISPRa screening community, and we recommend that this manuscript be accepted for publication.

Minor points:

- 1) There is a typo in Figure 1b in the plot title, which we think should be "Transcriptional Activation."
- 2) In Extended Data Fig. 2, please clarify what each point represents.
- 3) In line 128, the phrase "trend toward correlation" is probably best omitted. We suggest providing the range of correlation coefficients observed in the dual-promoter format instead.
- 4) Relating to point 3, we recommend that, within the text, the authors address possible causes for the decoupling of GFP expression and endogenous gene activation. Have the authors tried delivering the SA-sgRNA and the gene-targeting sgRNA on the same vector? One could imagine that differences in SA-sgRNA and gene-targeting sgRNA activity could be driven by variability in promoter strength, selection pressures and/or genomic integration sites. This could be added to the Discussion.
- 5) In the Results section titled "Optimized, multi-format synthetic guide RNAs for transient CRISPRa," a main figure is not cited until the last paragraph. We recommend making Extended Data Fig. 6 a main figure, because it is important for understanding this section.
- 6) Please list the catalog numbers for all antibodies and fluorophores used for flow cytometry in the supplementary methods table.
- 7) Please include a timeline in the main text that indicates when each component was delivered and when gene expression levels were measured by q-PCR and flow cytometry. This will indicate to the reader how long it takes for selection and activation to occur in this system.
- 8) To increase the accessibility of the work, please include an annotated version of the most important vectors (eg. Genbank file or marked up Word document); the current supplementary methods table is a bit difficult to parse.

Reviewer #2:

Remarks to the Author:

A versatile, high-efficiency platform for CRISPR-based gene activation

In this manuscript, the authors present a series of improvement to CRISPR-based gene activation using rational design. Broadly, three aspects of gene activation with a dCas9 system are optimized: promoter choice to maximize Cas9 expression, guide RNA design and an improved selection of functional cells upon introduction of the Cas9 embodiment.

Two improvements stand out in comparison to currently used CRISPRa systems. The self-selecting

sgRNA system to simplify CRISPRa cell line generation, a major bottleneck for many CRISPRa experiments, and the improvements to the MS2-harboring tetraloop of CRISPR SAM guide RNAs. Even though both improvements are quite specific to this CRISPR system, the underlying concepts could be applicable to other gene perturbation efforts.

Overall, the manuscript is clearly written, and the data shown convincingly support the stated claims.

- The manuscript inherently asks one important biological question. What is the functional fate of the self-activating sgRNA that remains expressed in the cells on the CRISPRa efficiency? As depicted in Fig 1a, the authors thoughtfully placed LoxP sites around the self-activating sgRNA and accompanying selection markers to eventually excise these. Actually removing the self-activating sgRNA after selection (via Cre) would address this question—and what is the impact on the CRISPRa activity. It's likely that CRISPR activity is reduced due to a loss of the activation permissive chromatin state. At the same time, the absence of the self-activating sgRNA might allow the Cas9 to load more other guide RNAs and therefore positively impact gene activation. In this light, a simple Cre treatment could answer the above question.

- There is no explanation of the components of the self-activating promoter and the guide RNA targeting it. This should be included.

- Along the above lines, the authors could address the question of whether using a weak sgRNA should inherently select for cells with more effective CRISPRa.

Minor points

- It would help to point out the lengths of the spacer part of the synthesized two-part guide RNA. This is the variable part to be synthesized when targeting different genes.

- Scales in panels displaying the percentage of cell with gene activation vary. Please scale them all to 100% so that the variability of efficiencies are more comparable. Similarly, MFI scales should be comparable at least within panels (eg. three different CD2 levels in Fig4c/e).

- Missing axes labels in Fig 3b,

Reviewer #3:
None

Reviewer #4:
None

We thank the reviewers for their thoughtful critiques and insightful comments. We have made every effort to address their concerns as detailed below. As a result, we believe we are now providing a significantly-improved manuscript compared to our original submission.

Point-by-point responses to reviewer comments and questions

Reviewer #1

1) There is a typo in Figure 1b in the plot title, which we think should be “Transcriptional Activation.”

We thank the reviewer for identifying this typo. It has been corrected in the revised figure.

2) In Extended Data Fig. 2, please clarify what each point represents.

We agree with the reviewer that our figure could be more effectively described. Therefore, we propose modifying the figure legend for Extended Data Fig. 2 in the following manner (new text underlined):

Original Text:

Extended Data Fig. 2: An integrated CRISPRa dependent GFP reporter is an inconsistent marker of CRISPRa efficiency across multiple cell lines.

a, Flow cytometric analysis of GFP CRISPRa reporter vs endogenous PD-L1 target gene activation across three CRISPRa piggyBac formats in three cell lines (Fig. 1). **b**, GFP CRISPRa reporter vs endogenous CD2 activation in three cell lines engineered with a CRISPRa-sel piggyBac. **c**, Flow cytometric analysis of GFP CRISPRa reporter vs two endogenous CRISPRa target genes (PD-L1, CXCR4) in clones derived from CRISPRa-sel populations pre-sorted on GFP expression using flow assisted cell sorting (FACS) in four cell lines. Bar graphs of GFP median fluorescence intensity (MFI) in clones normalized to parental cell line (left). Target gene expression in engineered clones infected with an endogenous gene targeting sgRNAs (PD-L1 or CXCR4) and normalized to non-targeting control gRNA (middle/right). **d**, Scatter plots showing correlation of normalized MFI for CRISPRa dependent GFP reporter vs endogenous target gene activation. R squared for simple linear regression analysis indicated.

Proposed Text:

Extended Data Fig. 2: An integrated CRISPRa dependent GFP reporter is an inconsistent marker of CRISPRa efficiency across multiple cell lines.

a, Flow cytometric analysis of GFP CRISPRa reporter vs endogenous PD-L1 target gene activation across three CRISPRa piggyBac formats in three cell lines (Fig. 1). **Pearson correlation values (R) between the fluorescence intensity of GFP and PD-L1 are shown at the top right of each panel.** **b**, GFP CRISPRa reporter vs endogenous CD2 activation in three cell lines engineered with a CRISPRa-sel piggyBac. **Pearson correlation values between the fluorescence intensity of GFP and CD2 are shown at the top right of each panel.** **c**, Flow cytometric analysis of GFP CRISPRa reporter vs two endogenous CRISPRa target genes (PD-L1, CXCR4) **in clonal lines** derived from CRISPRa-sel populations pre-sorted on GFP expression using flow assisted cell sorting (FACS) in four distinct cell line backgrounds. Bar graphs of GFP median fluorescence intensity (MFI) in clones normalized to parental cell line (left). **Each data point represents the normalized MFI value of an individual clone.** Target gene expression in engineered clones infected with an endogenous gene targeting sgRNAs (PD-L1 or CXCR4) and normalized to non-targeting control gRNA (middle/right). **d**, Scatter plots showing correlation of normalized MFI of **GFP vs endogenous target gene MFI for each individual clone.** **R² for simple linear regression analysis indicated.**

3) In line 128, the phrase “trend toward correlation” is probably best omitted. We suggest providing the range of correlation coefficients observed in the dual-promoter format instead.

We appreciate the reviewer’s comment, and agree that a statistical analysis is more appropriate than our original description of the data. Accordingly, we now provide Pearson correlation values, which were calculated for the data shown in Extended Data Fig. 2a-b. Correlation values were included above each plot in Extended Data Fig. 2a-b, and we propose modifying the text in the following manner.

Original Text:

Although a trend towards correlation was observed for the dual promoter format (Extended Data Fig. 2a-left), there was high variability in the single transcript and CRISPRa-sel contexts (Extended Data Fig. 2a-center, right). To further evaluate the relationship between GFP expression and endogenous gene activation in the CRISPRa-sel context, we expanded our analysis to a second endogenous target gene (CD2) (Extended Data Fig. 2b) and observed similarly weak correlations.

Proposed Text:

While correlations were generally higher in the dual promoter and single transcript formats (R=0.56-0.84 and R=0.40-0.86, respectively), correlations between

endogenous gene activation and GFP reporter expression were poor in the CRISPRa-select format (R=0.27-0.39) (Extended Data Fig. 2a). To further evaluate the relationship between GFP expression and endogenous gene activation in the CRISPRa-select context, we expanded our analysis to a second endogenous target gene (CD2) (Extended Data Fig. 2b) and observed similarly weak correlations (**R=0.35-0.38**).

4) Relating to point 3, we recommend that, within the text, the authors address possible causes for the decoupling of GFP expression and endogenous gene activation. Have the authors tried delivering the SA-sgRNA and the gene-targeting sgRNA on the same vector? One could imagine that differences in SA-sgRNA and gene-targeting sgRNA activity could be driven by variability in promoter strength, selection pressures and/or genomic integration sites. This could be added to the Discussion.

We appreciate the reviewer's suggestions and have incorporated many of these concepts into a newly expanded discussion section.

Proposed Text:

In our initial experiments we observed that while antibiotic selection with our SA system enabled generation of relatively uniform populations of CRISPRa-competent cells, further enrichment based on high SA GFP reporter expression did not result in cell populations with correspondingly robust endogenous gene activation. There are several possible explanations for this finding. Variability in the expression of the endogenous gene-targeting guide RNA due to positional effects of vector insertion could result in the decoupling of endogenous gene activation and SA GFP reporter expression. Separately, the GFP reporter in the SA context, unlike that in the dual promoter and single transcript systems, is transcriptionally linked to *puro*^r. Since we initially selected with puromycin, it is possible that the GFP in this configuration is acting as a passive marker, the expression of which is determined primarily by the selective requirement of the co-transcribed *puro*^r and secondarily by the level of CRISPR-mediated gene activation.

5) In the Results section titled "Optimized, multi-format synthetic guide RNAs for transient CRISPRa," a main figure is not cited until the last paragraph. We recommend making Extended Data Fig. 6 a main figure, because it is important for understanding this section.

We understand the reviewer's point, and we have updated the figures such that Extended Data Fig. 6 is now Fig. 4, and the original Fig. 4 is now referred to as Fig. 5.

6) Please list the catalog numbers for all antibodies and fluorophores used for flow cytometry in the supplementary methods table.

This is a good point from the reviewer. We have now included these in the 'Antibodies' tab of the Supplemental Methods sheet.

7) Please include a timeline in the main text that indicates when each component was delivered and when gene expression levels were measured by q-PCR and flow cytometry. This will indicate to the reader how long it takes for selection and activation to occur in this system.

We agree with the reviewer that this section could benefit from additional information. To complement the experiment-specific details, which can be found in the figure legends, we propose:

A) updating the results section text in the following manner:

Original text:

We evaluated the relative efficiency of each selection strategy in the human K562 cell line. Following puromycin selection, the individual populations were infected with lentiviral vectors expressing SAM-compatible sgRNAs targeting the promoter proximal regions of five cell surface receptor genes (Extended Data Fig. 1a).

Proposed Text:

We evaluated the relative efficiency of each selection strategy in the human K562 cell line. Here, CRISPRa vectors were electroporated, and following a minimum 5-day expansion period populations were selected with puromycin for a minimum of 1 week to remove cells which had not integrated the piggyBac vectors. Selected populations were subsequently transduced with lentiviral vectors (Extended Data Fig. 1a) expressing SAM-compatible sgRNAs targeting the promoter-proximal regions of five cell surface receptor genes (PD-L1, CD14, CD2, PROM1/CD133, CXCR4). Gene expression analysis was performed a minimum of 10 days following the introduction of each sgRNA and 1 week in the presence of zeocin.

B) Updating the relevant methods sections to include these details (below):

Flow cytometry

Antibody staining performed using manufacturers' recommended protocols and described in supplemental methods. Data collection performed on BD FACS Celesta or

BD FACS Symphony machines and analyzed by FlowJo 2 10.8.0 (Becton, Dickenson & Co.). Gating strategy indicated in Extended Data Fig. 1c. Live cell populations were gated using FSC and SSC profiles. Where relevant, lentivirally transduced cells specifically were examined by gating on mTagBFP2 positive populations. If cell populations were selected to greater than >95% mTagBFP2 positive then this gating step was omitted for some analyses. Populations were defined by gates established as indicated with 2 parameter pseudocolor plots (Extended Data Fig. 1c or Extended Data Fig. 6b) with identical control cell lines expressing a non-targeting control guide RNA and stained/collected in parallel. **Lentivirally-transduced populations were analyzed a minimum of 10 days post-infection with the gene-targeting sgRNA. Experiment specific timing is described in the associated figure legends. Synthetic guide RNA experiments were analyzed at day 3 post-transfection or electroporation.**

qRT-PCR

Cell pellets for qRT-PCR analysis were collected 14 days post lentiviral infection with a gene-targeting sgRNA, 11 days of which were in the presence of zeocin to fully select for cells with genome-integrated guides. RNA extraction was performed with a Quick-RNA 96 well kit (Zymo). cDNA generation was performed with a high-capacity cDNA synthesis kit using random primers and RNase inhibitor (Thermo) following recommended protocols. Quantitative RT-PCR was performed with an ABI QuantStudio 7 Flex real time PCR system. Relative quantification/fold change ($2^{-\Delta\Delta CT}$) analysis was performed by QuantStudio software. A GAPDH control gene was used for normalization purposes.

8) To increase the accessibility of the work, please include an annotated version of the most important vectors (e.g., Genbank file or marked up Word document); the current supplementary methods table is a bit difficult to parse.

This is an excellent point raised by the reviewer. In the ‘Full vector sequences’ tab of the Supplementary Methods, we have now included a link to the plasmid maps and annotated sequences, which are accessible through Benchling (no login or user account required).

Reviewer #2

1) The manuscript inherently asks one important biological question. What is the functional fate of the self-activating sgRNA that remains expressed in the cells on the CRISPRa efficiency? As depicted in Fig 1a, the authors thoughtfully placed LoxP sites around the self-activating sgRNA and accompanying selection markers to eventually excise these. Actually removing the self-activating sgRNA after selection (via Cre) would address this question—and what is the impact on the CRISPRa activity. It’s likely that CRISPR activity is reduced due to a loss of the activation permissive chromatin state. At the same time, the absence of the self-activating sgRNA might allow the Cas9

to load more other guide RNAs and therefore positively impact gene activation. In this light, a simple Cre treatment could answer the above question.

The reviewer presents an interesting question, which we believe we have now addressed with additional experiments in-line with these recommendations. These results can be found in a new Extended Data Fig. 6 along with related changes to the main text (below):

Proposed Text:

Our design of the CRISPRa-sel vector includes loxP sites flanking the SA sgRNA/selection cassette to enable its removal. While the presence of the SA cassette is necessary to drive selection marker expression, the SA sgRNA may compete to some extent with gene-targeting sgRNAs for dCas9 binding. In theory, this could impact the overall efficiency of target activation. To better understand how the presence of the SA cassette influences the efficiency of endogenous gene activation, we performed a series of experiments in cell lines with the intact SA cassette or following its removal with Cre recombinase post-selection (Extended Data Fig. 6). To this end, stable CAG-CRISPRa-sel K562 or 293T cell lines were transduced with gene-targeting sgRNAs (PD-L1, CD2, or CD14) (Extended Data Fig. 6a) and selected with zeocin to remove uninfected cells. Populations were then divided and either maintained in culture (mock) or treated with an mRNA encoding Cre. After 7 days, cells were evaluated by flow cytometry. Loss of the SA cassette was confirmed by the significant reduction in GFP expression in the Cre-treated populations (Extended Data Fig. 6b). Under these conditions no consistent difference in target gene activation was observed between the Cre-treated or untreated populations, suggesting competition from the SA sgRNA has minimal impact on target gene activation (Extended Data Fig. 6c-d). Furthermore, these data suggest that in short term studies, the SA cassette may not be essential for maintaining CRISPRa populations after the initial selection process.

2) There is no explanation of the components of the self-activating promoter and the guide RNA targeting it. This should be included.

We appreciate that the reviewer pointed out this oversight, and we agree that this is important information. We propose: A) adding the following text to describe the SA system in greater detail.

Original Text:

As a readout for CRISPRa function we also incorporated a GFP reporter downstream of a self-activating (SA) promoter, which could be activated only in the presence of functional CRISPRa machinery and a co-expressed SA-targeting guide RNA. Building on the self-activating concept, we devised a third strategy, which we term *CRISPRa selection* (CRISPRa-sel), where the puor^r gene is driven by the self-activating promoter and linked to the GFP reporter (Fig. 1a-bottom row).

Proposed Text:

*As a readout for CRISPRa function we also incorporated a GFP reporter downstream of a self-activating (SA) promoter sequence. **Here, a minimal promoter was derived from the TRE3G promoter (Takara Bio), in which the tet operator array has been replaced by a single 19bp sequence in context with a GGG protospacer-adjacent motif (PAM) in order to create a Cas9 targeting site. This target site can be recognized by a constitutive, H1 promoter-driven, self-activating sgRNA (SA sgRNA) that was designed to avoid association with endogenous human or mouse genomic sequences. Accordingly, in the presence of functional CRISPRa machinery, the SA sgRNA will activate expression of the co-encoded GFP reporter.** Building on the SA concept, we devised a third strategy, which we term CRISPRa selection (CRISPRa-sel), where the SA promoter is configured to express a T2A-linked puor^r and GFP reporter cassette (Fig. 1a-bottom row, **Extended Data Fig. 1a**).*

B) We have additionally generated a new Extended Data Fig. 1a which includes a more detailed schematic of the SA selection cassette.

3) Along the above lines, the authors could address the question of whether using a weak sgRNA should inherently select for cells with more effective CRISPRa.

We thank the reviewer for this interesting suggestion and have incorporated this idea in the new expanded discussion section.

Proposed text:

Manipulation of the SA guide RNA and/or its expression may provide yet another avenue for future optimization. In our system, the SA guide expression context (H1 promoter combined with the 2.0 sgRNA scaffold) is sub-optimal relative to the higher-efficiency U6 promoter and GNE-3 sgRNA employed for gene-targeting guides [Goguen et al. 2021 Mol. Therapy Nucleic Acids]. This may enable selection of more active CRISPRa populations, as only those cells which can

activate the puro^r gene under such limiting conditions will survive. If this hypothesis is correct, then it is possible that reducing the functionality of our SA sgRNA further, for example, by employing a scaffold that lacks MS2 aptamers or uses a less efficient spacer sequence, could result in more robust selective pressure and, consequently, improved activity within the selected population.

Minor points

- It would help to point out the lengths of the spacer part of the synthesized two-part guide RNA. This is the variable part to be synthesized when targeting different genes.

This is a good point from the reviewer. We have now added the text “Spacer 19-20nts” next to the guide RNA diagrams of the following figures: Fig. 4, Fig. 5, Extended Data Fig. 3 and Extended Data Fig. 8.

- Scales in panels displaying the percentage of cell with gene activation vary. Please scale them all to 100% so that the variability of efficiencies are more comparable. Similarly, MFI scales should be comparable at least within panels (e.g. three different CD2 levels in Fig4c/e).

We appreciate the reviewer’s comment on axis scaling, and we have attempted to address this with newly-revised figures. For the majority of figures, graphs representing % gene activation have now been scaled uniformly to 100%, and graphs depicting MFI have been unified on a per gene basis within their respective panels. During the course of generating these new graphs, however, it became clear that several graphs (5 total) necessitated the use of the original scaling strategy for clarity and interpretation.

Generally, this was restricted to contexts with suboptimal activation where it was visually challenging to discern statistically-significant and reproducible differences between conditions. Because the intent of these experiments is to compare across activator formats rather than between spacer sequences, we believe it is justified to use dynamic scaling for these rare examples, and we have indicated the dynamic axes in the appropriate figure legends.

Panels with author-recommended dynamic scaling:

- 1) Extended Data Fig. 1d (PC9 -PROM1 and CXCR4 %)***
- 2) Extended Data Fig. 1e (293T - PROM1 %)***

Dynamic scale (author recommended)

Fixed scale (reviewer recommended)

3) Extended Data Fig. 4b (KDR %+)

Dynamic scale (author recommended)

Fixed scale (reviewer recommended)

4) Extended Data Fig. 7 (CD2 gRNA-5 MFI)

Dynamic scale (author recommended)

Fixed scale (reviewer recommended)

5) Extended Data Fig. 9d (CXCR4 gRNA-1 MFI)

Dynamic Scale (author recommended)

Fixed scale (reviewer recommended)

- Missing axes labels in Fig 3b.

The labeling on our original figure was difficult to interpret. We have revised the figure, such that cell line labels have been moved from above the flow cytometry plots to below the plots for clarity.

Reviewers' Comments:

Reviewer #1:

Remarks to the Author:

The authors were responsive to all suggestions, and this paper is certainly ready for publication. If only all papers were as easy to review as this one!

Reviewer #2:

Remarks to the Author:

The authors have adequately responded to the critiques.